# Optogenetic Generation of Neural Firing Patterns with Temporal Shaping of Light Pulses

Himanshu Bansal [ID], Gur Pyari and Sukhdev Roy *[ID]

Department of Physics and Computer Science, Dayalbagh Educational Institute, Agra 282005, India; himanshubansal@dei.ac.in (H.B.); gurpyarirajput@dei.ac.in (G.P.)
* Correspondence: sukhdevroy@dei.ac.in

**Abstract:** The fundamental process of information processing and memory formation in the brain is associated with complex neuron firing patterns, which can occur spontaneously or be triggered by sensory inputs. Optogenetics has revolutionized neuroscience by enabling precise manipulation of neuronal activity patterns in specified neural populations using light. However, the light pulses used in optogenetics have been primarily restricted to square waveforms. Here, we present a detailed theoretical analysis of the temporal shaping of light pulses in optogenetic excitation of hippocampal neurons and neocortical fast-spiking interneurons expressed with ultrafast (Chronos), fast (ChR2), and slow (ChRmine) channelrhodopsins. Optogenetic excitation has been studied with light pulses of different temporal shapes that include square, forward-/backward ramps, triangular, left-/right-triangular, Gaussian, left-/right-Gaussian, positive-sinusoidal, and left-/right-positive sinusoidal. Different light shapes result in significantly different photocurrent amplitudes and kinetics, spike-timing, and spontaneous firing rate. For short duration stimulations, left-Gaussian pulse results in larger photocurrent in ChR2 and Chronos than square pulse of the same energy density. Time to peak photocurrent in each opsin is minimum at right-Gaussian pulse. The optimal pulse width to achieve peak photocurrent for non-square pulses is 10 ms for Chronos, and 50 ms for ChR2 and ChRmine. The pulse energy to evoke spike in hippocampal neurons can be minimized on choosing square pulse with Chronos, Gaussian pulse with ChR2, and positive-sinusoidal pulse with ChRmine. The results demonstrate that non-square waveforms generate more naturalistic spiking patterns compared to traditional square pulses. These findings provide valuable insights for the development of new optogenetic strategies to better simulate and manipulate neural activity patterns in the brain, with the potential to improve our understanding of cognitive processes and the treatment of neurological disorders.

**Keywords:** optogenetics; temporal shaping; ChRmine; Chronos; channelrhodopsin





## 1. Introduction

The intricate spatiotemporal patterns of neuronal firing are the basis of information processing, memory formation, and numerous neurological disorders in the brain [1–3]. Manipulating or disrupting these firing patterns using neurostimulation techniques has become a common approach for understanding their underlying mechanisms and developing treatments for neurological diseases such as Parkinson's disease and epilepsy, and brain–computer interfaces [4–7]. Optogenetics has revolutionized neuroscience by enabling precise manipulation of neuronal activity patterns in specified neural populations using light [8–10]. It has a wide range of applications in and beyond neuroscience that include heart, peripheral, and retina [11–15]. However, the conventional method of optogenetic stimulation typically relies on square-shaped pulse trains, which may not reflect the natural patterns of neural activity. Recent research suggests that employing irregular temporal patterns of optogenetic stimulation can produce more naturalistic activity patterns, potentially leading to more effective therapeutic outcomes in a range of neuropsychiatric applications [16–19].

Light-sensitive proteins are essential tools in optogenetics that offer a diverse range of properties, including light sensitivity, photocurrent kinetics and amplitude, ion selectivity, and mechanisms of ion transportation [20,21]. Channelrhodopsin-2 (ChR2) is one of the most extensively studied opsins in optogenetics, but its use is limited, as it requires high irradiances to evoke spikes, leading to temperature changes after sustained illumination [22,23]. In addition, it results in multi-spiking in interneurons due to the slow turn-off kinetics (turn-off time ~12 ms) of its photocurrent [24,25]. Chronos, a blue and green light-sensitive ultrafast opsin (turn-off time ~3.5 ms) that has been discovered, can overcome these limitations, as it exhibits higher sensitivity and can evoke spikes with sub-millisecond temporal precision [26,27]. In parallel, opsins with red-shifted activation wavelengths have become important due to their suitability for deep excitation, as red-light can penetrate deeper into the brain tissue in comparison to blue and green [28]. Most recently, a new pump—like channelrhodopsin, namely ChRmine—was discovered. Although the turn-off kinetics of the photocurrent in ChRmine (turn-off time ~50 ms) is relatively slow, it exhibits several orders of magnitude higher photocurrent and light sensitivity at 590 nm [29,30]. Despite the different applications for these opsins, it remains unclear how they will respond to light pulses of different temporal shapes. Further investigations are necessary to determine the effect of temporal shaping of light pulses on the photocurrent amplitude, kinetics, and spike timing in the optogenetic excitation of neurons expressing these different opsins.

The impact of temporally shaped light pulses in optogenetic excitation of the hippocampus has provided new insights on memory and learning. To date, only a few studies have reported the effect of optical stimulation with different pulse patterns on neural spiking [16,18,19]. The different pulse patterns, including sinusoidal pulse, square pulse, ramp pulse, forward-ramp pulse, backward-ramp pulse and Gaussian pulse, are used for various applications in different light-sensitive opsin-expressing neurons [16,18,19]. Recently, different waveforms have been used in CA3 and CA1 for memory replay extension and synaptic weight formulation [19]. Rhythms in neural activity are observed across various temporal and spatial scales and are referred to as oscillations. Earlier studies have shown that these oscillations play an important role in neural communication, computation, and cognition [17,31–33]. A comparative experimental study of different stimulation patterns suggested that pulsed and sinusoidal stimulations induce highly synchronous spiking with higher variability over the stimulation period [34].

Heating is an important issue while delivering high-intensity light pulses for long durations [35,36]. In most of the experiments, square-shaped pulses are used to evoke spikes using opsins with different turn-on and -off photocurrent kinetics. The kinetics of photocurrents can be significantly changed on changing the temporal shape of the illuminating pulse. Hence, it is essential to uncover the effect of light shape in order to maximize the photocurrent at minimum pulse energy.

The application of computational modelling in the field of optogenetics has significantly enhanced our understanding of the intricate dynamics involved in the generation of photocurrents within the opsin molecule and the subsequent spiking in opsin-expressing neurons [9]. In recent years, various studies have delved into the computational modelling of optogenetic systems, with initial efforts aimed at developing accurate models for the ChR2 photocycle and light-to-spike conversion [37,38]. Another vital aspect that has been examined in this area is to illuminate sub-cellular compartments and activation in scattering tissue mediums [39–41]. In addition to these efforts, recent theoretical modelling studies have reported the use of light-driven chloride pumps and channels for optogenetic inhibition, excitation, and bidirectional control of different types of neurons [42–48]. These investigations have led to a deeper understanding of the factors that impact the efficiency and specificity of optogenetic tools in various biological contexts. Overall, computational modelling in optogenetics has provided new avenues for exploring the complex interplay between light and biology, paving the way for innovative approaches to manipulate and understand biological processes.

The selection of light pulses is important in optogenetic interrogation of the cellular interactions and their contribution to coordinated activity patterns in neuronal networks. However, there is no comprehensive study comparing the effect of different pulse patterns on different light-sensitive opsins expressing neurons. The objective of this paper was to theoretically study the impact of different temporal shaped pulses on the photocurrent kinetics of ultrafast, fast and slow channelrhodopsins, excitation of hippocampal neurons and fast-spiking neocortical interneurons and find optimized temporal shapes for different applications.

## 2. Methods

All channelrhodopsins used in optogenetics sense light through the embedded retinal molecule. Light-triggered photoisomerization in the opsin-bound retinal molecule results in conformational changes in opsin structure that result in the opening of an ion-conducting pore across the membrane [49–53]. Thus, ions flow across the membrane and result in depolarization or hyperpolarization of membrane potential. Figure 1 shows a schematic of an integrated biophysical model of optogenetic control of opsin-expressing neurons and different temporal shapes of illuminating light pulses used in this study.

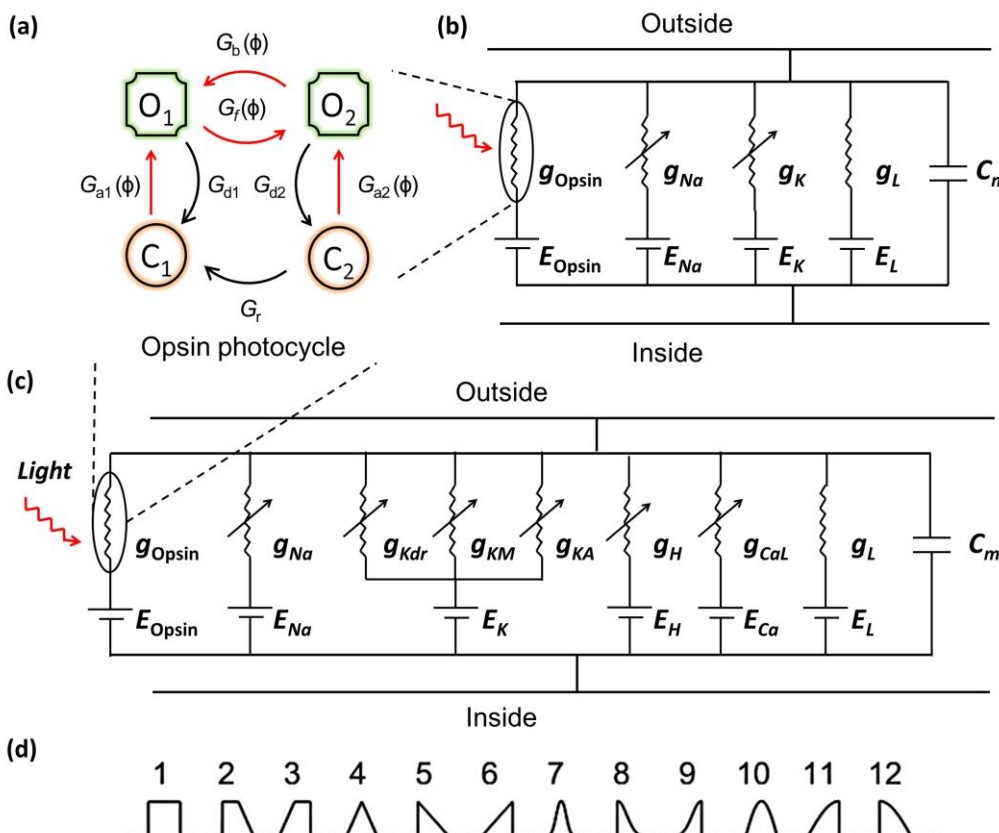

**Figure 1.** Biophysical circuit model for optogenetic control of neurons and illuminating pulse shapes. (**a**) 4-state model of opsin photocurrent. (**b**,**c**) Schematic of equivalent circuit diagram of opsin-expressing (**b**) fast-spiking neocortical interneurons, and (**c**) hippocampal neurons. (**d**) Different temporal shapes of illuminating light pulses, 1: Square, 2: Forward-Ramp, 3: Backward-Ramp, 4: Triangular, 5: Right-Triangular, 6: Left-Triangular, 7: Gaussian, 8: Right-Gaussian, 9: Left-Gaussian, 10: Positive-Sinusoidal, 11: Left-Positive-Sinusoidal, and 12: Right-Positive-Sinusoidal [40,43,47,54].

### 2.1. Photocurrent Model

The light-induced ionic current (photocurrent) through the opsin channels ($I_{Opsin}$) can be expressed as,

$$I_{Opsin} = g_{Opsin} f_\varphi(\varphi, t)(V - E_{Opsin}) \tag{1}$$

where $g_{Opsin}$ is the total conductance, $f_\varphi(\varphi, t)$ is a normalized light-dependent function that accounts for instantaneous population density of conducting state at time ($t$) and depends on photon flux per unit area per unit time ($\varphi$), $V$ is the neuron membrane potential, and $E_{Opsin}$ is reversal potential for opsin channel [43,54]. $\varphi$ is defined as $\lambda I / hc$, where $h$ is Planck's constant, $I$ is irradiance, $\lambda$ is wavelength, and $c$ is the speed of light in vacuum [45,54].

Reported experiments have shown that ChR2, Chronos, and ChRmine exhibit a biphasic decay of their photocurrent, which needs at least two open states with different turn-off kinetics to accurately describe the kinetics [24,26,29]. Thus, a four-state model with two non-conducting closed $C_1$ and $C_2$ and two conducting open states $O_1$ and $O_2$ has been considered for all three opsins (Figure 1a) [45,46]. In dark, the opsin molecule rests in the closed ground state $C_1$. On absorption of photons, the opsin excites to the first open state $O_1$. From $O_1$, it either decays to the second open state $O_2$, which is less conductive but has a longer lifetime, or to $C_1$. Similarly, from $O_2$, it either transits back to $O_1$ or decays to the second closed state $C_2$. The reversible transitions between $O_1$ and $O_2$ states are both light- and thermal-induced. From closed state $C_2$, it either thermally relaxes to closed state $C_1$, which is a very slow process called recovery, or is photo-excited back to $O_2$ [45,46,54].

If $C_1, O_1, C_2,$ and $O_2$ denote instantaneous fractions of population density in each of the four states, the rate of change can be described by the following set of differential equations,

$$\dot{C}_1 = G_{d1}O_1 + G_r C_2 - G_{a1}(\phi)C_1 \tag{2}$$

$$\dot{O}_1 = G_{a1}(\phi)C_1 + G_b(\phi)O_2 - \left(G_{d1} + G_f(\phi)\right)O_1 \tag{3}$$

$$\dot{O}_2 = G_{a2}(\phi)C_2 + G_f(\phi)O_1 - (G_{d2} + G_b(\phi))O_2 \tag{4}$$

$$\dot{C}_2 = G_{d2}O_2 - (G_r + G_{a2}(\phi))C_2 \tag{5}$$

where $C_1 + O_1 + O_2 + C_2 = 1$. $G_{a1}, G_{a2}, G_{d1}, G_{d2}, G_f, G_b$ and $G_r$ are the rate constants for transitions $C_1 \to O_1$, $C_2 \to O_2$, $O_1 \to C_1$, $O_2 \to C_2$, $O_1 \to O_2$, $O_2 \to O_1$ and $C_2 \to C_1$, respectively, determined from experimental data and defined as $G_{a1}(\phi) = k_1 \phi^p / (\phi^p + \phi_m^p)$, $G_{a2}(\phi) = k_2 \phi^p / (\phi^p + \phi_m^p)$, $G_f(\phi) = G_{f0} + k_f \phi^q / (\phi^q + \phi_m^q)$, $G_b(\phi) = G_{b0} + k_b \phi^q / (\phi^q + \phi_m^q)$ [43,54]. Since two open states are considered in the four-state model, $f_\phi(\phi, t) = O_1 + \gamma O_2$, where $\gamma = g_{O2}/g_{O1}$. $g_{01}$ and $g_{02}$ are the conductances of states $O_1$ and $O_2$, respectively [43,54]. The values of model parameters were determined from reported experimental results (Table 1) [24,26,29,45,46,54].

**Table 1.** Parameters for ChR2, Chronos and ChRmine [24,26,29,45,46,54].

| Parameter | Chronos | ChR2 | ChRmine |
|---|---|---|---|
| $G_{d1}$ (ms$^{-1}$) | 0.278 | 0.09 | 0.02 |
| $G_{d2}$ (ms$^{-1}$) | 0.01 | 0.01 | 0.013 |
| $G_r$ (ms$^{-1}$) | $1.2 \times 10^{-3}$ | $0.5 \times 10^{-3}$ | $5.9 \times 10^{-4}$ |
| $g_0$ (nS) for photocurrent | 39 | 5.9 | 110 |
| $g_0$ (mS/cm$^2$) for hippocampal neurons | 0.88 | 0.65 | 1.9 |
| $g_0$ (mS/cm$^2$) for neocortical interneurons | 0.176 | 0.12 | 0.38 |
| $\Phi_m$ (ph·mm$^{-2}$·s$^{-1}$) | $8 \times 10^{15}$ | $4 \times 10^{16}$ | $2.1 \times 10^{15}$ |
| $k_1$ (ms$^{-1}$) | 1.8 | 3 | 0.2 |
| $k_2$ (ms$^{-1}$) | 0.01 | 0.18 | 0.01 |
| $G_{f0}$ (ms$^{-1}$) | 0.05 | 0.015 | 0.0027 |
| $G_{b0}$ (ms$^{-1}$) | 0.08 | 0.005 | 0.0005 |
| $k_f$ (ms$^{-1}$) | 0.1 | 0.03 | 0.001 |
| $k_b$ (ms$^{-1}$) | 0.01 | 0.003 | 0 |
| $\gamma$ | 0.05 | 0.05 | 0.05 |
| $p$ | 0.8 | 1 | 0.8 |
| $q$ | 0.9 | 1 | 1 |
| $\lambda$ (nm) | 470 | 470 | 590 |
| $E$ (mV) | 0 | 0 | 5.64 |

## 2.2. Model for Optogenetic Excitation of Opsin-Expressing Neurons

To compute the change in membrane potential due to the ionic photocurrent through the expressed opsin molecules, an integrated model of optogenetic control was formulated by integrating the photocurrent into the biophysical models, namely the Wang–Buzaski interneuron model (for fast-spiking neocortical interneurons) and Hemond neuron model (for hippocampal neurons) (Figure 1b,c) [55–57].

The rate of change in membrane potential in these opsin-expressing neurons and interneurons can be expressed as follows:

$$C_m \dot{V} = -I_{ionic} + I_{DC} + I_{opsin} \tag{6}$$

where $C_m$ is the membrane capacitance, $I_{DC}$ is the constant direct electric current that controls the excitability. $I_{ionic}$ is a sum of natural voltage-gated ionic currents through different ion channels embedded within the neuron membrane.

For hippocampal neurons, $I_{ionic}$ is defined as,

$$I_{ionic} = I_{Na} + I_{Kdr} + I_H + I_{CaL} + I_{KA} + I_{KM} + I_L \tag{7}$$

where each ionic current was modelled as $I_f = g_f m^p h^q \left( V - E_f \right)$, where $g_f$ is the maximum conductance, $m$ and $h$ are the activation (with exponent $p$) and inactivation gating variables, respectively, and $E_f$ is the reversal potential, except $I_L = g_L(V - E_L)$. Each gating function $x$ ($m$ or $h$) obeys the first-order kinetics as $\dot{x} = (x_\infty - x)/\tau_x$, where the $x_\infty$ and $\tau_x$ are the voltage-dependent functions given in Table 2 [46,55,56]. Other model parameters are given in Table 3 [46,55,56].

**Table 2.** Gating function parameters of ion channels in Hemond neuron circuit model [46,55,56].

| $I_{ionic}$ | Gating Variable | $\alpha$ | $\beta$ | $x_\infty$ | $\tau_x$ (ms) |
|---|---|---|---|---|---|
| $I_{Na}$ | $p = 3$ | $\dfrac{-0.4(V+6)}{\exp\left[-\left(\frac{V+6}{7.2}\right)\right]-1}$ | $\dfrac{0.124(V+6)}{\exp\left[\frac{V+6}{7.2}\right]-1}$ | $\dfrac{\alpha}{\alpha+\beta}$ | $\dfrac{0.4665}{\alpha+\beta}$ |
| | $q = 1$ | $\dfrac{-0.03(V+21)}{\exp\left[-\frac{(V+21)}{1.5}\right]-1}$ | $\dfrac{0.01(V+21)}{\exp\left[\frac{V+21}{1.5}\right]-1}$ | $\dfrac{1}{\exp\left[\frac{V+21}{4}\right]+1}$ | $\dfrac{0.4662}{\alpha+\beta}$ |
| $I_{Kdr}$ | $p = 1$ | $\exp[-0.113(V-37)]$ | $\exp[-0.0791(V-37)]$ | $\dfrac{1}{1+\alpha}$ | $\dfrac{50*\beta}{1+\alpha}$ |
| $I_H$ | $q = 1$ | $\exp[0.0833(V+75)]$ | $\exp[0.0333(V+75)]$ | $\dfrac{1}{\exp\left[\frac{V+73}{8}\right]+1}$ | $\dfrac{\beta}{0.0575(1+\alpha)}$ |
| $I_{CaL}$ | $p = 2$ | $\dfrac{15.69(-V+81.5)}{\exp\left[\frac{-V+81.5}{10}\right]-1}$ | $0.29*\exp\left(-\frac{V}{10.86}\right)$ | $\dfrac{\alpha}{\alpha+\beta}$ | $\dfrac{2*\exp(0.00756(V-4))}{1+\exp(0.0756(V-4))}$ |
| $I_{KA}$ | $p = 1$ | $\exp\left[-0.0564(V-35)-\dfrac{0.0376(V-35)}{\exp\left(\frac{V+16}{5}\right)+1}\right]$ | $\exp\left[-0.0315(V-35)-\dfrac{0.021(V-35)}{\left(\exp\left(\frac{(V+16)}{5}\right)+1\right)}\right]$ | $\dfrac{1}{1+\alpha}$ | $\dfrac{3.045*\beta}{1+\beta}$ |
| | $q = 1$ | $\exp[0.0113(V+32)]$ | - | $\dfrac{1}{1+\alpha}$ | $0.26(V+26)$ |
| $I_{KM}$ | $p = 1$ | $\dfrac{0.016}{\exp\left[\frac{-(V+52.7)}{23}\right]}$ | $\dfrac{0.016}{\exp\left[\frac{V+52.7}{18.8}\right]}$ | $\dfrac{1}{\exp\left[\frac{-(V+16)}{10}\right]+1}$ | $60+\dfrac{\beta}{0.003(1+\alpha)}$ |

**Table 3.** Hemond neuron model parameters [46,55,56].

| Parameter | Unit | Value |
|---|---|---|
| $g_{Na}$ | mS/cm$^2$ | 22 |
| $g_{Kdr}$ | mS/cm$^2$ | 10 |
| $g_H$ | mS/cm$^2$ | 0.01 |
| $g_{CaL}$ | mS/cm$^2$ | 0.01 |
| $g_{KA}$ | mS/cm$^2$ | 20 |
| $g_{KM}$ | mS/cm$^2$ | 0.5 |
| $g_L$ | mS/cm$^2$ | 0.04 |
| $E_H$ | mV | $-30$ |
| $E_{Na}$ | mV | 55 |
| $E_K$ | mV | $-90$ |
| $E_L$ | mV | $-70$ |
| $\tau_{Ca}$ | ms | 100 |
| $I_{DC}$ | μA/cm$^2$ | 0 |
| $C_m$ | μF/cm$^2$ | 1.41 |

For fast-spiking neocortical interneurons, $I_{ionic}$ can be expressed as,

$$I_{ionic} = I_{Na} + I_K + I_L \tag{8}$$

where $I_{Na} = g_{Na}m_\infty^3 h(V - E_{Na})$, $I_K = g_K n^4(V - E_K)$, and $I_L = g_L(V - E_L)$. $g_{Na}$, $g_K$ and $g_L$ are the maximal conductance, and $E_{Na}$, $E_K$ and $E_L$ are the reversal potential for the sodium, potassium and leakage ionic currents, respectively [57]. $h$ and $m_\infty$ are inactivation and activation variables for sodium current, respectively, and n is the inactivation variable for potassium current. The gating variable $x$ ($n$ or $h$) obeys the first-order kinetics, $\dot{h} = \eta[\alpha_h(1-h) - \beta_h h]$, $\dot{n} = \eta[\alpha_n(1-n) - \beta_n n]$, and $m_\infty = \alpha_m/(\alpha_m + \beta_m)$ (Tables 4 and 5) [45,57]. All of the simulations were performed using the fourth-order Runge–Kutta method implemented in MATLAB R2019b.

**Table 4.** Gating function parameters of ion channels in Wang–Buzsaki interneuron circuit model [45,57].

| $I_{ionic}$ | Gating Variable | $\alpha$ | $\beta$ |
|---|---|---|---|
| $I_{Na}$ | $p = 3$ | $\frac{-0.1(V+35)}{\exp\left[-\left(\frac{V+35}{10}\right)\right]-1}$ | $4 * \exp\left[\frac{-(V+60)}{18}\right]$ |
| | $q = 1$ | $0.07 * \exp\left[-\left(\frac{V+58}{20}\right)\right]$ | $\frac{1}{\exp\left[\frac{V+28}{10}\right]+1}$ |
| $I_K$ | $p = 4$ | $\frac{-0.01 * (V+34)}{\exp\left[\frac{-(V+34)}{10}\right]-1}$ | $0.125 * \exp\left[-\left(\frac{V+44}{80}\right)\right]$ |

**Table 5.** Wang–Buzsaki interneuron model parameters [45,57].

| Parameter | Unit | Value |
|---|---|---|
| $g_{Na}$ | mS/cm$^2$ | 35 |
| $g_K$ | mS/cm$^2$ | 9 |
| $g_L$ | mS/cm$^2$ | 0.1 |
| $E_{Na}$ | mV | 55 |
| $E_K$ | mV | $-90$ |
| $E_L$ | mV | $-65$ |
| $\eta$ | - | 7 |
| $I_{DC}$ | μA/cm$^2$ | $-0.51$ |
| $C_m$ | μF/cm$^2$ | 1.41 |
| $V_{rest}$ | mV | $-65$ |

*2.3. Temporal Shapes of Light Pulses*

Different temporal shapes of illuminating light pulses are shown in Figure 1d. Mathematical expressions of different temporal shapes of light pulses are given in Table 6.

**Table 6.** Temporal shapes of light pulses for optogenetic excitation of opsin-expressing neurons. $t$ is time. $t_1$ and $t_2$ are the times at which the light pulse is turned on and off, respectively [58–60].

| Shape Name | Shape | Mathematical Expression for Pulse Amplitude ($I$) |
|---|---|---|
| Square pulse |  | $I(t) = \begin{cases} 1, for\ t_1 < t < t_2 \\ 0, \quad otherwise \end{cases}$ |
| Forward-Ramp |  | $I(t) = \begin{cases} 1, & t_1 < t < \frac{t_2+t_1}{2} \\ (t_2 - t)/\left[\frac{t_2-t_1}{2}\right], & \frac{t_2+t_1}{2} < t < t_2 \\ 0, & otherwise \end{cases}$ |
| Backward-Ramp |  | $I(t) = \begin{cases} (t - t_1)/\left[\frac{t_2-t_1}{2}\right], for\ t_1 < t < \frac{t_2+t_1}{2} \\ 1, \qquad for\ \frac{t_2+t_1}{2} < t < t_2 \\ 0, \qquad\qquad otherwise \end{cases}$ |

**Table 6.** *Cont.*

| Shape Name | Shape | Mathematical Expression for Pulse Amplitude (*I*) |
|---|---|---|
| Triangular |  | $I(t) = \begin{cases} (t-t_1)/\left[\frac{t_2-t_1}{2}\right], & t_1 < t < \frac{t_2-t_1}{2} \\ (t_2-t)/\left[\frac{t_2-t_1}{2}\right], & \frac{t_2-t_1}{2} < t < t_2 \\ 0, & otherwise \end{cases}$ |
| Left-Triangular |  | $I(t) = \begin{cases} \frac{t-t_1}{t_2-t_1}, & for\ t_1 < t < t_2 \\ 0, & otherwise \end{cases}$ |
| Right-Triangular |  | $I(t) = \begin{cases} 0, & otherwise \\ \frac{t_2-t}{t_2-t_1}, & for\ t_1 < t < t_2 \end{cases}$ |
| Gaussian |  | $I(t) = \begin{cases} \exp\left(-\frac{(t-c)^2}{2\sigma^2}\right), & for\ t_1 < t < t_2 \\ 0, & otherwise \end{cases}; \sigma = \frac{(t_2-t_1)/2}{3.5}; \\ c = (t_2+t_1)/2$ |
| Left-Gaussian |  | $I(t) = \begin{cases} \exp\left(-\frac{(t-c)^2}{2\sigma^2}\right), & for\ t_1 < t < t_2 \\ 0, & otherwise \end{cases}; \sigma = \frac{(t_2-t_1)}{3.5}; c = t_2$ |
| Right-Gaussian |  | $I(t) = \begin{cases} \exp\left(-\frac{(t-c)^2}{2\sigma^2}\right), & for\ t_1 < t < t_2 \\ 0, & otherwise \end{cases}; \sigma = \frac{(t_2-t_1)}{3.5}; c = t_1$ |
| Positive-Sinusoidal |  | $I(t) = \begin{cases} \sin\left(\frac{\pi(t-t_1)}{t_2-t_1}\right), & for\ t_1 < t < t_2 \\ 0, & otherwise \end{cases}$ |
| Left-Positive-Sinusoidal |  | $I(t) = \begin{cases} \sin\left(\frac{(\pi/2)(t-t_1)}{t_2-t_1}\right), & for\ t_1 < t < t_2 \\ 0, & otherwise \end{cases}$ |
| Right-Positive-Sinusoidal |  | $I(t) = \begin{cases} \sin\left(\frac{(\pi/2)(t_2-t)}{t_1-t_2}\right), & for\ t_1 < t < t_2 \\ 0, & otherwise \end{cases}$ |

## 3. Results

The effect of temporal changes in light pulse shapes on photocurrent in different opsins on illuminating with short (5 ms) and long (1 s) light pulses has been studied in detail (Figure 2). The amplitude of light pulses has been considered to have two different conditions: (i) iso-max (pulses having the same maximum irradiance), and (ii) iso-energy density (pulses having the same energy density). For generating iso-max pulses, the amplitude of non-square light pulses is kept the same as the amplitude of the square pulse. For generating iso-energy density pulses, non-square light pulses of higher irradiances are used to match the area under the curve between pulse irradiance and time. The pulse irradiance for iso-energy density pulses is 1 mW/mm$^2$ for square, 1.336 mW/mm$^2$ for forward-/backward-ramp, 2 mW/mm$^2$ for triangular/left-triangular/right-triangular, 2.635 mW/mm$^2$ for Gaussian/left-Gaussian/right-Gaussian, and 1.575 mW/mm$^2$ for positive-sinusoidal/left-positive-sinusoidal/right-positive-sinusoidal pulse. The energy density at these irradiances is 5 μJ/mm$^2$ for a 5 ms light pulse and 1 mJ/mm$^2$ for a 1 s light pulse. The wavelength of light is 470 nm for Chronos and ChR2 and 590 nm for ChRmine.

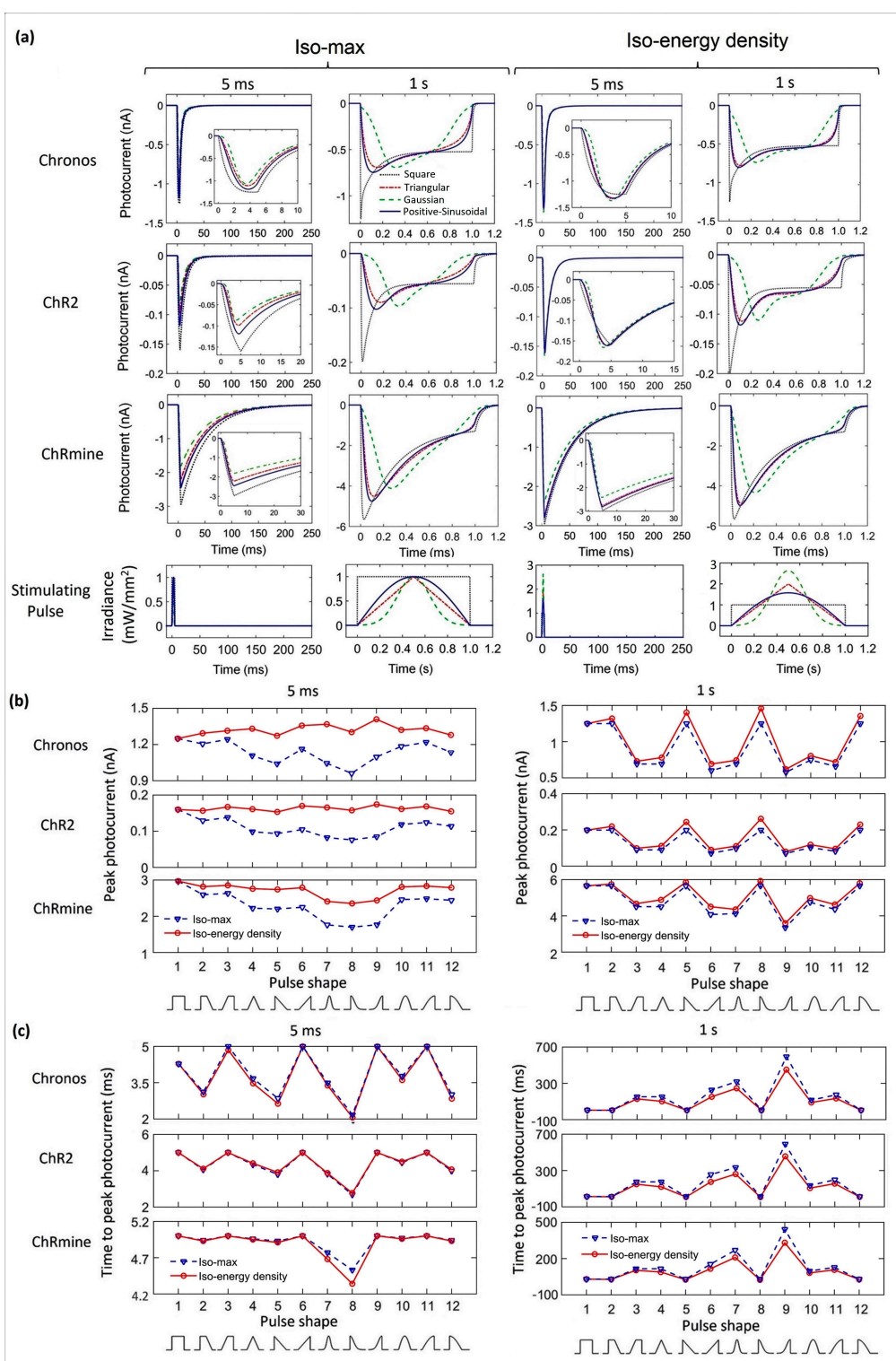

**Figure 2.** Photocurrent in ultrafast (Chronos), fast (ChR2), and slow (ChRmine) channelrhodopsins-expressing neurons under patch clamp at −70 mV on illuminating with light pulses of different temporal shapes at 470 nm for ChR2 and Chronos, and 590 nm for ChRmine. (**a**) Variation of photocurrent with time on illuminating with 5 ms and 1 s light pulses of different shapes (left) at same peak irradiance 1 mW/mm² (iso-max), and (right) at same energy density 5 μJ/mm² (for 5 ms pulse) and 1 mJ/mm² (for 1 s light pulse) (iso-energy density). Inset shows corresponding zoomed-in variation of photocurrent with time at 5 ms pulse. The lower panel shows the variation of illuminating

pulse irradiance with time. (**b**,**c**) Corresponding variation of (**b**) peak photocurrent, and (**c**) time to achieve peak photocurrent with shapes of illuminating light pulses of (left) 5 ms and (right) 1 s under both iso-max and iso-energy density conditions. The numbers on the x-axis indicate different pulses, 1: Square, 2: Forward-Ramp, 3: Backward-Ramp, 4: Triangular, 5: Right-Triangular, 6: Left-Triangular, 7: Gaussian, 8: Right-Gaussian, 9: Left-Gaussian, 10: Positive-Sinusoidal, 11: Left-Positive-Sinusoidal, and 12: Right-Positive-Sinusoidal.

The photocurrent in different opsins at short and long light pulses is shown in Figure 2a. Corresponding values of peak photocurrent and time to peak at square, triangular, Gaussian and positive-sinusoidal pulse shapes are given in Table 7. At short iso-energy density pulses, the peak photocurrent maximally increases by 0.12 nA in Chronos and 0.005 nA in ChR2, whereas it decreases by 0.56 nA in ChRmine on changing pulse shape from square to Gaussian. At short iso-max pulses, the peak photocurrent decreases maximally on changing the pulse shape from square to Gaussian as 0.18 nA in Chronos, 0.08 nA in ChR2 and 1.15 nA in ChRmine. At short light pulses, time to peak photocurrent is longest for square pulse and shortest for Gaussian pulse in all three opsins under both iso-max and iso-energy density pulses. It maximally decreases by 1.67 ms in Chronos, 1.08 ms in ChR2, and 0.32 ms in ChRmine on changing the pulse shape from square to Gaussian (Table 7). At a long light pulse (1 s), the photocurrent kinetics significantly change on changing the pulse shape (Table 7). Time to peak photocurrent maximally increases by 305.7 ms in Chronos, 317 ms in ChR2, and 230.8 ms in ChRmine on changing the illuminating pulse shape from square to Gaussian (Figure 2a). This is due to the slow turn-on of Gaussian pulses. As the change in photocurrent kinetics significantly changes the spike latency and evoked firing patterns, pulse shaping can be used as an additional means to generate different firing patterns with a single opsin. On illuminating with 1 s light pulse, the photocurrent at Gaussian pulse turns off slowly due to the slow turn-off of Gaussian light pulse, unlike the square pulse, which shows a stable plateau (Figure 2a). Since the firing rate of neurons corresponds to the amplitude of the input photocurrent, the turn-off kinetics of the firing rate evoked by the Gaussian pulse would be smoother compared to that by the square pulse.

The variation of peak photocurrent and time to reach peak photocurrent at different temporal shapes and their subtypes is shown in Figure 2b,c. The peak photocurrent in all three opsins at short light pulses is slightly smaller for non-square light pulses at iso-max, while at iso-energy density pulses, it is larger for some non-square light shaped pulses in ChR2 and Chronos (Figure 2b). At short light pulses, under iso-max condition, the peak photocurrent is minimum at right-Gaussian light pulse in all three opsins, while it is maximum in ChR2 and Chronos at left-Gaussian shaped light pulse under iso-energy density condition (Figure 2b). On changing the pulse shape from square to left-Gaussian pulse at the same energy density, the peak photocurrent increases from 1.24 nA to 1.41 nA in Chronos, and 0.16 nA to 0.174 nA in ChR2, respectively.

At longer light pulses, there is no significant difference in the peak photocurrent on switching between iso-max and iso-energy density conditions (Figure 2b). However, the change in shape results in a much larger difference in peak photocurrent in all three opsins. Under long duration pulses, the peak current is maximum for right-Gaussian pulses in all three opsins (Figure 2b).

For evoking precise spiking, it is important to determine which pulse shape can evoke a large photocurrent with fast turn-on kinetics. The variation of time to reach peak photocurrent at different shapes of light pulse is shown in Figure 2c, which is almost similar for both iso-max and iso-energy density pulses but significantly varies on changing the pulse shape. At short light pulses, the fastest turn-on of photocurrent is achieved in all three opsins on illuminating with right-Gaussian light pulse, at which the peak photocurrent in Chronos and ChR2 is almost similar to the square pulse under the iso-energy density condition (Figure 2c).

**Table 7.** Effect of pulse shapes on peak photocurrent and time to peak in Chronos, ChR2 and ChRmine at short (5 ms) and long (1 s) light pulse at 470 nm for Chronos and ChR2, and 590 nm for ChRmine under iso-max and iso-energy density conditions.

| Pulse Shape | | | Square | | Triangular | | Gaussian | | Positive-Sinusoidal | |
|---|---|---|---|---|---|---|---|---|---|---|
| Condition | Irradiance/ Energy Density | Pulse Width | $I_{peak}$ (nA) | $t_{peak}$ (ms) | $I_{peak}$ (nA) | $t_{peak}$ (ms) | $I_{peak}$ (nA) | $t_{peak}$ (ms) | $I_{peak}$ (nA) | $t_{peak}$ (ms) |
| | | | | | Chronos | | | | | |
| Iso-max | 1 mW/mm$^2$ | 5 ms | 1.24 | 5 | 1.10 | 3.7 | 1.06 | 3.6 | 1.18 | 3.9 |
| | 1 mW/mm$^2$ | 1 s | 1.25 | 4.3 | 0.68 | 160 | 0.69 | 310 | 0.74 | 130 |
| Iso-energy density | 5 μJ/mm$^2$ | 5 ms | 1.24 | 5 | 1.33 | 3.4 | 1.36 | 3.33 | 1.32 | 3.5 |
| | 1 mJ/mm$^2$ | 1 s | 1.24 | 4.3 | 0.778 | 110 | 0.74 | 250 | 0.803 | 80 |
| | | | | | ChR2 | | | | | |
| Iso-max | 1 mW/mm$^2$ | 5 ms | 0.16 | 5 | 0.098 | 4.42 | 0.08 | 3.92 | 0.11 | 4.5 |
| | 1 mW/mm$^2$ | 1 s | 0.19 | 13 | 0.089 | 160 | 0.096 | 330 | 0.1 | 140 |
| Iso-energy density | 5 μJ/mm$^2$ | 5 ms | 0.160 | 5 | 0.161 | 4.5 | 0.165 | 4.01 | 0.161 | 4.5 |
| | 1 mJ/mm$^2$ | 1 s | 0.19 | 13.6 | 0.111 | 110 | 0.110 | 260 | 0.118 | 100 |
| | | | | | ChRmine | | | | | |
| Iso-max | 1 mW/mm$^2$ | 5 ms | 2.97 | 5 | 2.21 | 4.94 | 1.82 | 4.77 | 2.45 | 4.96 |
| | 1 mW/mm$^2$ | 1 s | 5.65 | 29.2 | 4.50 | 120 | 4.11 | 260 | 4.74 | 100 |
| Iso-energy density | 5 μJ/mm$^2$ | 5 ms | 2.97 | 5 | 2.76 | 4.92 | 2.41 | 4.68 | 2.8 | 4.93 |
| | 1 mJ/mm$^2$ | 1 s | 5.65 | 28.1 | 4.88 | 90 | 4.34 | 210 | 4.98 | 80 |

The effect of change in light shapes on opsin photocurrent at different irradiances, pulse widths and pulse frequencies was studied in detail (Figure 3). Variation of peak photocurrent with irradiance on illuminating with 1 s light is shown in Figure 3a. It is evident that for lower irradiance < 0.01 mW/mm$^2$, the peak current in all three opsins for square pulses is almost similar to the photocurrent for other shapes. However, above 0.01 mW/mm$^2$, the difference significantly increases with irradiance (Figure 3a). Further, the effect of change in pulse width on peak photocurrent is shown in Figure 3b. The effect of change in pulse width has been studied at effective power density for 50% activation (EPD50), which is 0.28 mW/mm$^2$ for Chronos, 0.65 mW/mm$^2$ for ChR2, and 0.04 mW/mm$^2$ for ChRmine. Interestingly, the peak photocurrent saturates above a pulse width for the square pulse and decreases with pulse width above a threshold for non-square shaped pulses. The pulse width at which maximum photocurrent can be generated is 10 ms for Chronos, and 50 ms for ChR2 and ChRmine (Figure 3b).

The return to baseline is an important factor to determine temporal resolution to evoke spikes through multiple stimulations. In Figure 3c, the variation of return to baseline (RTB) (%) with pulse stimulation frequency at different pulse shapes is shown. The RTB % is lowest for the square-shaped pulse, whereas it is highest for the Gaussian-shaped pulse in all three opsins (Figure 3c).

Light-evoked spiking in hippocampal neurons at different shapes of pulses is shown in Figure 4. It is important to know the irradiance thresholds corresponding to each light shape. The minimum irradiance threshold (MIT) to evoke action potential in different opsin-expressing neurons changes on changing the shape of the light pulse due to differences in amplitude and kinetics of the generated photocurrent. On illuminating with 5 ms light pulse, MIT at different pulse shapes is shown in Figure 4a. MIT is lowest for square-shaped pulses and highest for left-/right-Gaussian pulses in all three opsins. On changing the pulse shape of the light pulse from square to left- or right-Gaussian, the MIT increases from 0.19 mW/mm$^2$ to 0.75 mW/mm$^2$ with Chronos, 0.58 mW/mm$^2$ to 1.52 mW/mm$^2$ with ChR2, and 27 μW/mm$^2$ to 91 μW/mm$^2$ for ChRmine, respectively (Figure 4a).

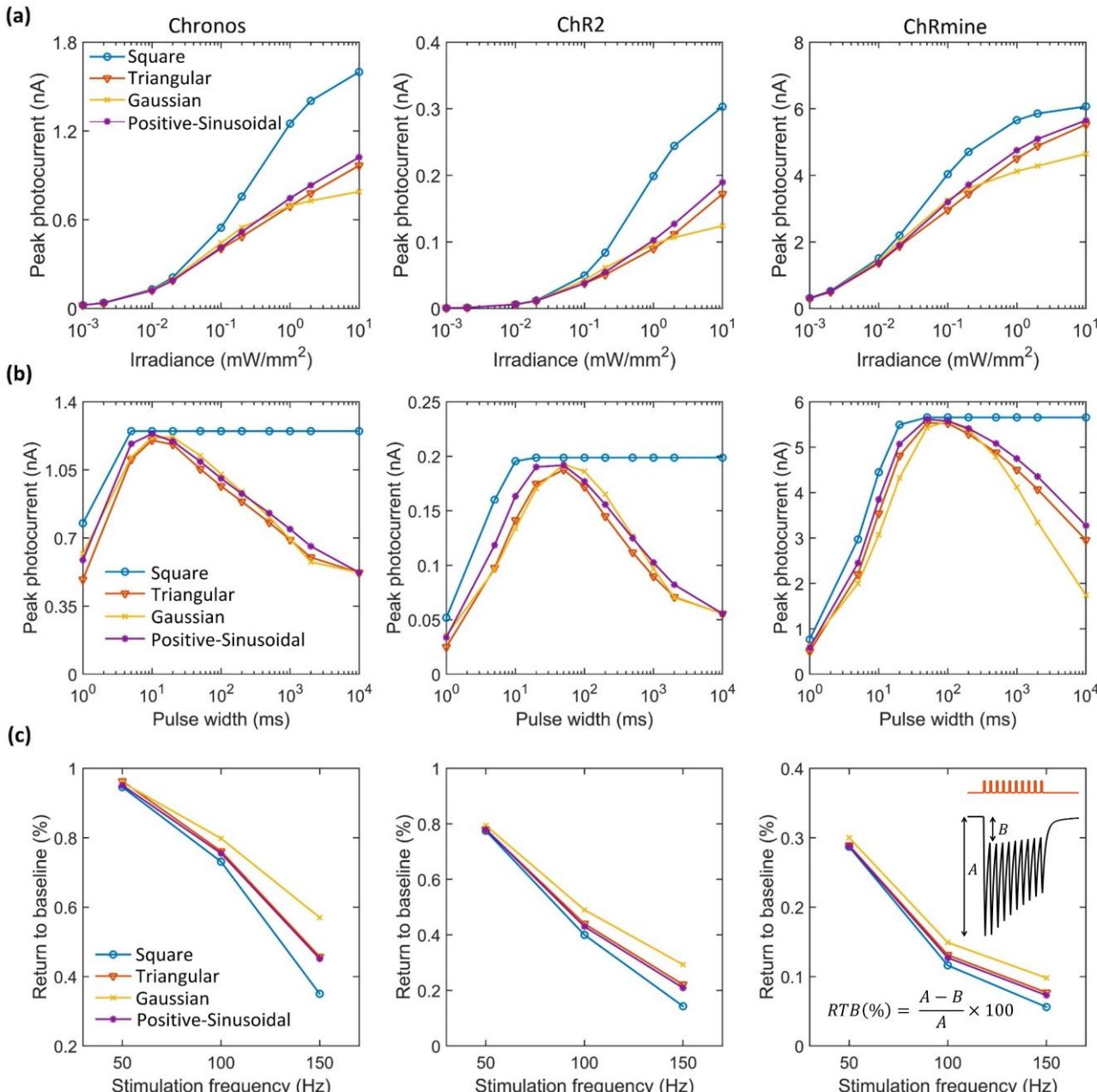

**Figure 3.** Effect of irradiance, pulse width and stimulation frequency on photocurrent evoked by light pulses of different shapes in Chronos, ChR2, and ChRmine at 470 nm for ChR2 and Chronos, and 590 nm for ChRmine. (**a**,**b**) Variation of peak photocurrent with (**a**) irradiance on illuminating with 1 s light pulse, and (**b**) pulse width on illuminating at effective power density for 50% activation (EPD50) corresponding to each opsin (0.28 mW/mm$^2$ for Chronos, 0.65 mW/mm$^2$ for ChR2, and 0.04 mW/mm$^2$ for ChRmine. (**c**) Variation of return to baseline ratio (RTB) (%) with stimulation frequency on illuminating with multiple 5 ms light pulses at 1 mW/mm$^2$. Inset: Formula for calculating RTB (%).

Figure 4b shows the corresponding minimum energy of light pulses at different pulse shapes to evoke spikes. It is evident that the shape of the light pulse with the lowest energy is square for Chronos, Gaussian for ChR2, and positive-sinusoidal or left-/right-positive sinusoidal for ChRmine (Figure 4b). The minimum pulse energy decreases from 2.9 μJ/mm$^2$ to 2.54 μJ/mm$^2$ on changing the pulse from square to Gaussian with ChR2,

and 0.13 $\mu J/mm^2$ to 0.1 $\mu J/mm^2$ with ChRmine on changing the pulse from square to positive-sinusoidal, respectively (Figure 4b).

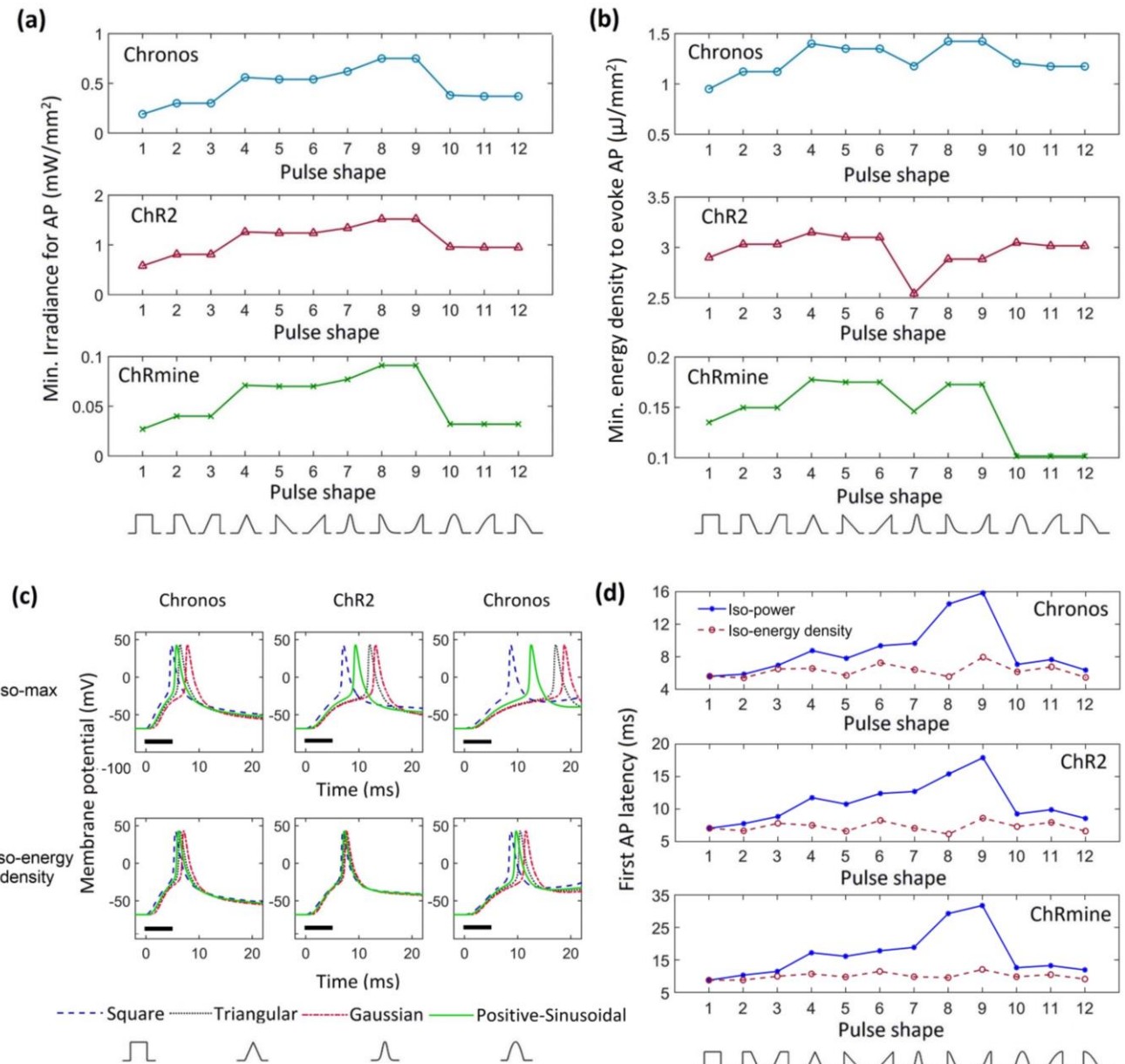

**Figure 4.** Optogenetic excitation of Chronos-, ChR2- and ChRmine-expressing hippocampal neurons using light pulses of different temporal shapes at 470 nm for ChR2 and Chronos, and 590 nm for ChRmine. (**a**) Variation of minimum irradiance threshold to evoke action potential (AP) on illuminating with 5 ms light pulse of indicated shapes. (**b**) Variation of normalized minimum energy density of 5 ms light to evoke AP with pulse shape. (**c**) Variation of membrane potential with time on illuminating with 5 ms light pulse of different shapes with iso-max at 0.75 $mW/mm^2$ for Chronos, 1.56 $mW/mm^2$ for ChR2, and 0.091 $mW/mm^2$ for ChRmine, and with iso-energy density at 3.75 $\mu J/mm^2$ for Chronos, 7.8 $\mu J/mm^2$ for ChR2, and 0.455 $\mu J/mm^2$ for ChRmine. (**d**) Variation of first AP latency with light shape on illuminating with 5 ms light pulse with iso-max amplitude and iso-energy density similar to (**c**). The numbers on the x-axis in (**a**,**b**,**d**) indicate pulses of different shapes, 1: Square, 2: Forward-Ramp, 3: Backward-Ramp, 4: Triangular, 5: Right-Triangular, 6: Left-Triangular, 7: Gaussian, 8: Right-Gaussian, 9: Left-Gaussian, 10: Positive-Sinusoidal, 11: Left-Positive-Sinusoidal, and 12: Right-Positive-Sinusoidal.

Spike timing is an important factor in information processing in the brain that plays a crucial role in synaptic plasticity. The light-evoked spikes at different light shapes under iso-max and iso-energy density conditions are shown in Figure 4c. It is evident that different pulse shapes result in different spike latencies. The variation of spike latency at different pulse shapes at iso-max and iso-energy density with each opsin is shown in Figure 4d. The spike latency does not change significantly under the iso-energy density condition, while it significantly changes in the iso-max condition. The most delayed spike is generated with a left-Gaussian-shaped pulse for all three opsins (Figure 4d). The analysis reveals that spikes of latencies ranging from 5.6 ms to 15.85 ms with Chronos, 7.05 ms to 17.9 ms with ChR2, and 8.75 ms to 31.75 ms with ChRmine can be generated through the studied temporal light shapes (Figure 4d).

Optogenetic excitation of fast-spiking neocortical interneurons was analyzed to study the instantaneous variation of firing rate evoked through different shapes of light pulses (Figure 5). Figure 5a shows the variation of instantaneous firing rate with time on illuminating with pulses of different shapes. For light pulse shapes in which the pulse amplitude decreases with time, the firing rates decrease in all three opsins. However, for light shapes in which the pulse amplitude increases with time, the firing rates increase up to a certain maximum frequency in all three opsins and subsequently saturate in ChRmine and decrease in Chronos and ChR2. The decrease in firing rate in Chronos and ChR2 is due to fast desensitization of their photocurrent (Figure 5a). The timing of maximum firing at each pulse shape is shown in Figure 5b. It is evident that ChR2 and Chronos-expressing interneurons achieve their maximum firing rates significantly before the maximum light pulse, and the difference increases for pulses with delayed maxima. In ChRmine, the firing rate reaches its maxima after the maxima of the light pulse only when illuminated with light pulses with fast turn-on. Different combinations of these temporal shapes can help in generating naturalistic firing patterns, important to better decode information encoded in natural firing patterns evoked through sensory inputs or spontaneous activity.

Further, the effect of frequency of stimulating waveforms of different shapes on the firing rate in each opsin-expressing interneuron is shown in Figure 6. It is evident that the firing rate varies smoothly on illuminating with light of non-square waveforms, which is more naturalistic than illuminating with square waveforms (Figure 6a). In addition, the firing rate variation in ChRmine is smoother in comparison to other opsins (Figure 6a). It is due to the slowest turn-off kinetics of the photocurrent in ChRmine among the studied opsins. The analysis also reveals that the contrast ratio (ratio of maximum and minimum firing rates in each cycle of waveform) is lower for non-square pulses (Figure 6b). This is due to the slow turn-off of the opsin photocurrent at non-square pulses (Figure 2a). In comparison with ChRmine, ChR2 and Chronos exhibit better contrast at each type of waveform. Under sustained illumination, the contrast ratio decreases with time in Chronos and ChR2, whereas it increases in ChRmine (Figure 6b–d). The analysis also reveals that in each opsin, the triangular waveform exhibits the lowest contrast among the studied waveforms.

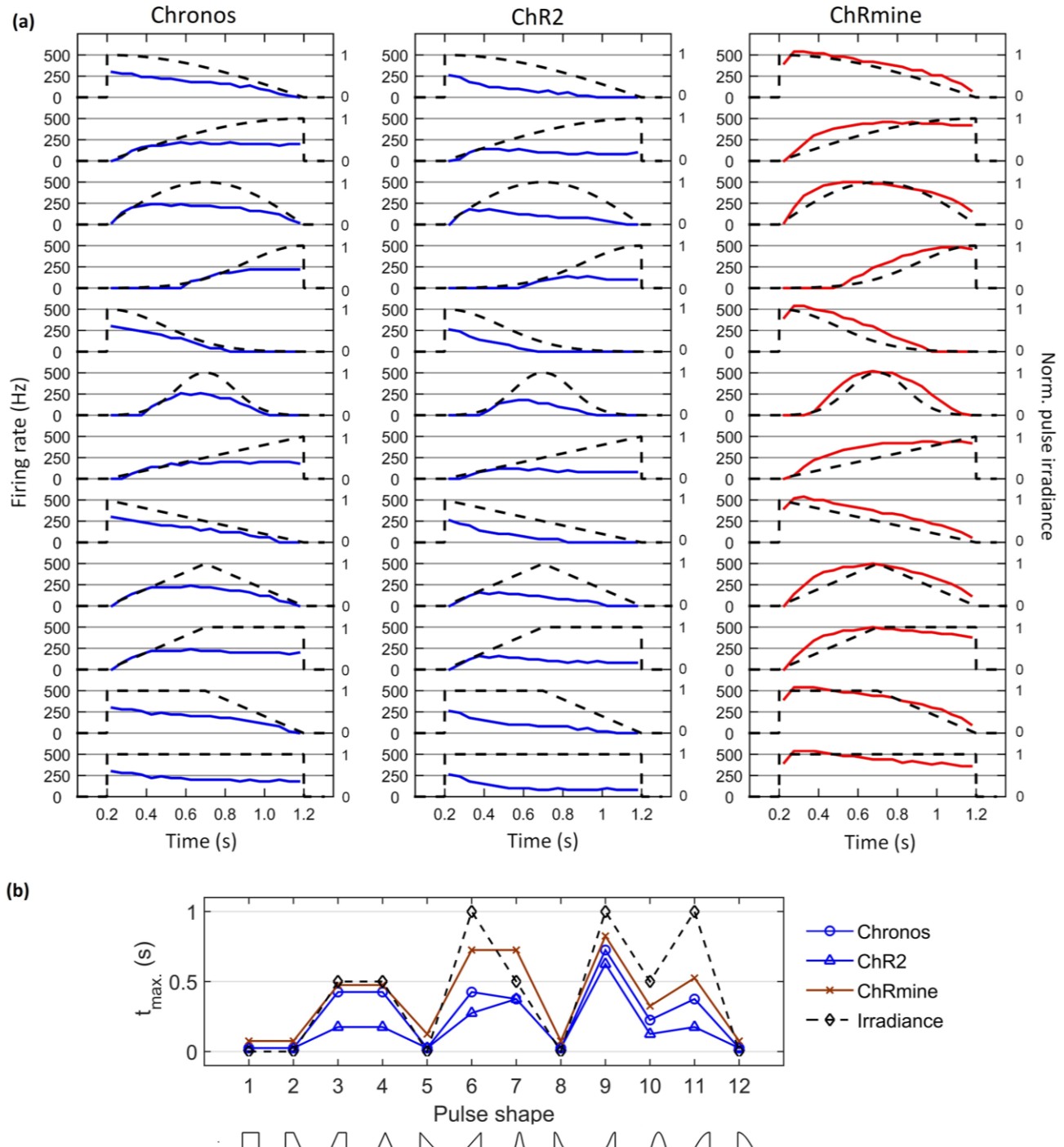

**Figure 5.** Instantaneous firing rate in neocortical interneurons expressed with Chronos, ChR2 and ChRmine on illuminating with 1 s light pulses of different temporal shapes at 470 nm for ChR2 and Chronos, and 590 nm for ChRmine. (**a**) Variation of instantaneous firing rate (blue solid line for ChR2 and Chronos and red solid line for ChRmine) and normalized pulse irradiance (black dashed line) with time on illuminating with light pulse of different temporal shapes at effective power density for 50% activation (EPD50), i.e., 0.28 mW/mm² for Chronos, 0.65 mW/mm² for ChR2, and 0.04 mW/mm² for ChRmine. (**b**) Corresponding variation of time to reach maximum firing rate (solid line) and time to reach maximum irradiance (dashed line) on illuminating with pulse shape. The numbers on the x-axis indicate pulses of different shapes, 1: Square, 2: Forward-Ramp, 3: Backward-Ramp, 4: Triangular, 5: Right-Triangular, 6: Left-Triangular, 7: Gaussian, 8: Right-Gaussian, 9: Left-Gaussian, 10: Positive-Sinusoidal, 11: Left-Positive-Sinusoidal, and 12: Right-Positive-Sinusoidal.



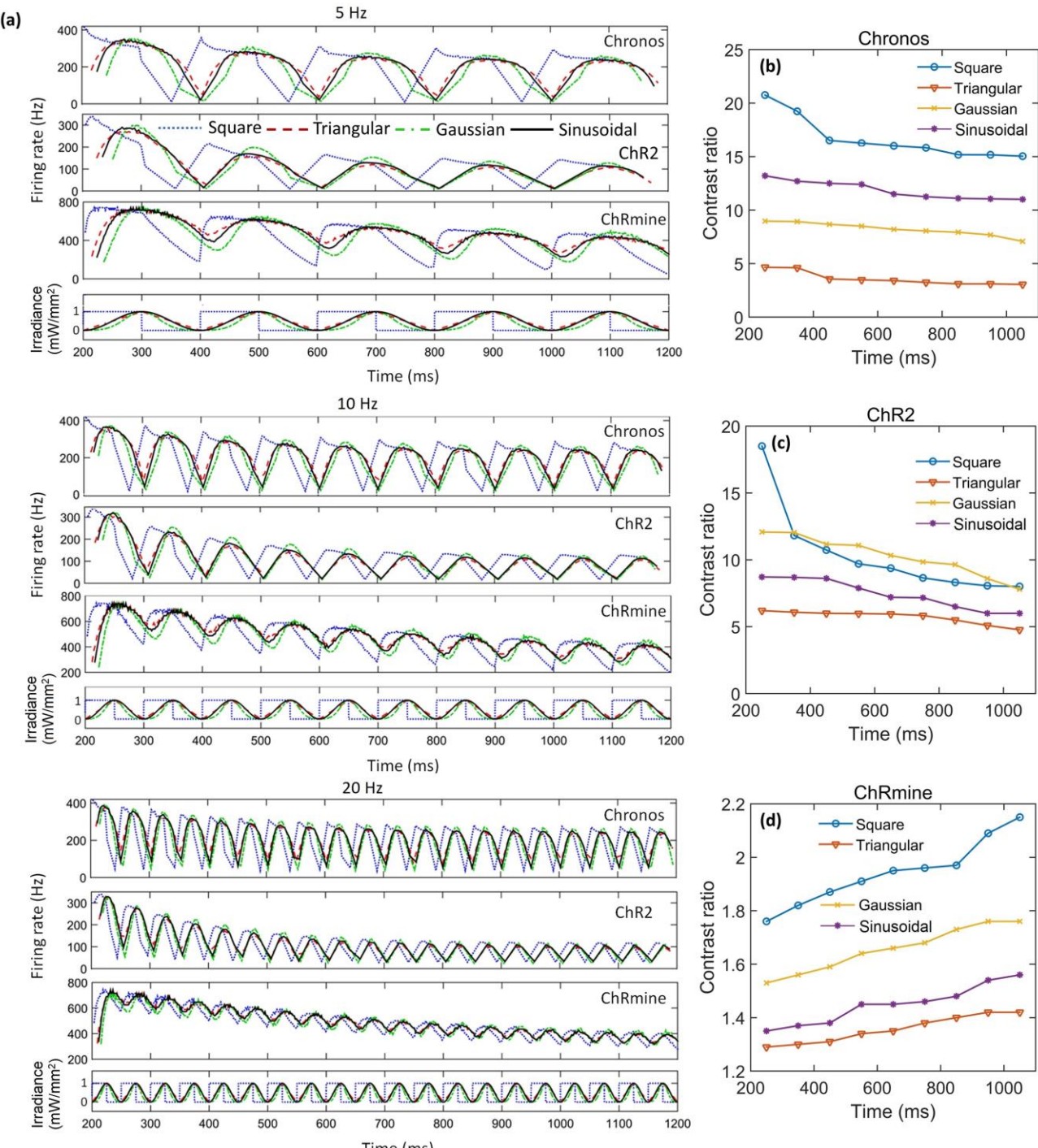

**Figure 6.** Instantaneous firing rate in neocortical interneurons expressed with Chronos, ChR2, and ChRmine on illuminating with square, triangular, Gaussian and sinusoidal waveforms at 5, 10 and 20 Hz at 470 nm for ChR2 and Chronos, and 590 nm for ChRmine. (**a**) Variation of instantaneous firing rate in interneurons on illuminating with iso-max waveforms of different shapes at frequency (upper) 5 Hz, (middle) 10 Hz, and (lower) 20 Hz at 1 mW/mm². Corresponding variation of applied waveform amplitudes with time is shown below each firing rate plot. (**b–d**) Corresponding variation of contrast ratio (ratio of maximum and minimum firing rate in each cycle of waveform) with time at 10 Hz.

## 4. Discussion

In this paper, the effect of temporally shaped light pulses in optogenetic excitation of hippocampal neurons and neocortical interneurons expressed with ultrafast, fast, and slow channelrhodopsins was studied in detail. Different temporal shapes of light pulses, including square, forward-/backward-ramp, triangular, left-/right-triangular, Gaussian, left-/right-Gaussian, positive-sinusoidal, and left-/right-positive sinusoidal, resulted in different types of firing patterns and action potentials with a wide range of latencies, which are useful to study various processes in the brain. The analysis revealed that spikes can be evoked at significantly lower energy than square light pulses for Gaussian-shaped pulses with ChR2 and positive-sinusoidal pulses with ChRmine.

Spike timing and firing rate play important roles in various neurological phenomena that include temporal spike coding, neural plasticity, pathology, spike timing-dependent plasticity associated with memory and learning, induction of synaptic long-term potentiation and depression, activity restoration in degenerated retina, auditory nerves, and cortical areas, and prosthesis [19,27,61,62]. In most of the experiments, stimulus-driven spikes were regular, and the relative timing between pre- and post-synaptic events was constant, whereas the actual firing patterns in the cortex of the intact animal was found to be irregular, and the timing between pre- and post-synaptic events varied [61,63]. In the present study, it was shown that different temporal shapes of light pulses result in different firing patterns (Figure 5). It was shown that by varying the temporal shape of the light pulse under the iso-max condition, the spike latency can be varied from 5.6 ms to 15.85 ms with Chronos, 7.05 ms to 17.9 ms with ChR2, and 8.75 ms to 31.75 ms with ChRmine (Figure 4). These results are very useful for generating arbitrary sequences of pre- and post-synaptic events with varying time differences.

Opsins with distinct kinetics generate different temporal patterns of evoked activity and differently regulate cortical gamma-oscillations [64]. It is reported that rapid onset kinetics of opsin photocurrent may facilitate the recruitment of highly precise initial spike responses, whereas slow onset kinetics preclude synchronous spiking and result in delayed peak responses [64]. The present study showed that temporal shaping of light pulses is an additional means to control turn-on and -off rates of the opsin photocurrent and corresponding firing rate (Figures 5 and 6).

Light-sensitive opsins exhibit a wide range of photocurrent kinetics and amplitudes and photosensitivity. Chronos, with its ability to generate precise spikes at higher frequencies, has enabled applications of optogenetics in temporal spike coding and the study of auditory systems [26,65]. ChRmine, a recently discovered pump-like cation-conducting channelrhodopsin, exhibits high sensitivity along with much larger photocurrents that create new opportunities [29,30,66]. ChRmine has been used for low-power, deep and large volume optogenetic excitation of neurons and has shown promising results in cardiac pacing and vision restoration [14,67]. Although these opsins confer the ability to optically mimic the full dynamic range of natural firing patterns without introducing artifacts, the study of their photocurrent kinetics has been primarily limited to square-shaped light pulses [25,26,29,68]. In this paper, a detailed comparison of photocurrents in ultrafast, fast and slow channelrhodopsins under different temporally shaped light pulses provided new insights. The study showed that on illuminating with similar pulse profiles, different opsins resulted in different firing patterns (Figures 5 and 6). Thus, the opsin kinetics play an additional role along with the pulse shape in generating firing patterns in optogenetics, which is an important feature to improve recently reported computational studies [18,19].

In optogenetics, light-driven chloride pumps are widely used for precise suppression of ongoing electrical activity in the targeted neurons [44,69–71]. These pumps include NpHR, a halorhodopsin from the archaeon *Natronomas pharaonis*, eNpHR2.0/3.0, modified versions of NpHR for enhanced membrane targeting, and Jaws, cruxhalorhodopsin extracted from *Haloarcula* (Halobacterium) *salinarum* (strain Shark) [69–71]. Sustained activation of these pumps changes the intracellular chloride concentration, which subsequently changes the amplitude and/or polarity of naturally occurring chloride channels. After

the turn-off of light, rebound of chloride results in post-illumination excitation [71,72]. To overcome the limitation, it has been reported that the post-illumination firing can be minimized using a ramp-like turn-off of the light pulse [45,71]. The present study showed that non-square waveforms result in slow turn-on and -off of photocurrent and firing rate, which would directly affect the rate of chloride rebound (Figure 2 and 5). Chloride channels suppressed the ongoing electrical activity of neurons on sustained illumination with light [73]. However, pulsed illumination results in excitation of electrical activity with these chloride channels due to more positive reversal potential than resting membrane potential [73]. A detailed theoretical study of the effect of different pulse shapes on the photocurrent kinetics in halorhodopsins and light-gated chloride channels would provide insights to efficiently suppress neuronal activity.

Heating is an important issue when sustained firing patterns are generated through optical excitation of a large neuronal population. It has been reported that commonly used photo-stimulation conditions change the targeted tissue temperature by 0.2–2 °C, sufficient to suppress spiking in multiple brain regions and affect various physiological processes inside the brain [35,36]. The study suggests that pulse duration and duty cycle play an important role in minimizing heating effects. In this paper, we have explicitly shown that fast opsins result in larger photocurrent when illuminated with left-Gaussian-shaped pulses of the same duration and same energy in comparison to square-shaped pulses (Figure 2b). Additionally, there is an optimal pulse width for non-square-shaped pulses to achieve maximum photocurrent, i.e., 10 ms for Chronos and 50 ms for ChR2 and ChRmine (Figure 3b). Furthermore, the study has shown that Gaussian-shaped light pulses with ChR2 and positive-sinusoidal-shaped light pulses with ChRmine evoke spikes in hippocampal neurons at much lower energy in comparison to square pulses (Figure 4b).

## 5. Conclusions

The present study of optogenetic excitation of ultra-fast, fast and slow channelrhodopsin-expressing neurons using square- and different non-square-shaped pulses allowed us to generate a wide range of spiking patterns, which are needed to mimic and investigate natural firing patterns in different brain regions. The detailed theoretical analysis explicitly showed how temporal shaping of light pulses changes photocurrent kinetics and amplitudes, spike latency, the minimum energy threshold to evoke spikes, firing pattern shapes, and contrast under sustained illumination. These findings provide valuable insights for the development of new optogenetic strategies to better simulate and manipulate neural activity patterns in the brain, with the potential to improve our understanding of cognitive processes and the treatment of neurological disorders.

The present study considered a four-state model of the opsin photocycle that accurately simulates the opsin photocurrent. However, the experimentally reported photocycles of these channelrhodopsins have shown additional intermediate states [74]. Although new models with a higher number of states will further enhance the accuracy, they would also result in increased complexity and computational time. The study also considered single compartment biophysical circuit models of neurons and interneurons. However, different neuronal compartments have different ion-channel compositions and density; hence, the use of multicompartmental models is better to simulate the optogenetic response of neurons [75].

The use of non-square light pulses with lower energy would be helpful in minimizing heating effects in targeted tissue. The generation of action potentials with different latencies by temporal shaping of light pulses would provide an additional means to study spike timing-dependent plasticity associated with memory and learning. The present theoretical study considered models of ChR2, Chronos and ChRmine, and four type of pulse shapes and their subtypes. Although the study provided significant information on how opsins with different kinetics respond to different pulse shapes, the extension of these models to other opsins and pulse shapes would lead to more insights. Multicompartmental and network level modelling of neurons would further extend the scope of study on the

effect of temporal pulse shaping on optogenetically evoked neuron firing patterns and network dynamics.

**Author Contributions:** Conceptualization, H.B., G.P. and S.R.; methodology, H.B., G.P. and S.R.; software, H.B. and G.P.; validation, H.B. and G.P.; formal analysis, H.B., G.P. and S.R.; investigation, H.B., G.P. and S.R.; resources, S.R.; data curation, H.B. and G.P.; writing—original draft preparation, H.B. and G.P.; writing—review and editing, S.R.; visualization, H.B. and G.P.; supervision, S.R.; project administration, S.R.; funding acquisition, S.R. All authors have read and agreed to the published version of the manuscript.

**Funding:** This work was supported by the University Grants Commission, India [F.530/14/DRS-III/2015(SAP-I)]; and Department of Science and Technology, India [CRG/2021/005139 and MTR/2021/000742 to S.R.; INSPIRE Fellowship to H.B. [DST/INSPIRE/03/2017/003087].; Junior Research Fellowship to G.P.].

**Institutional Review Board Statement:** Not applicable.

**Informed Consent Statement:** Not applicable.

**Data Availability Statement:** All data are presented in the manuscript and figures.

**Acknowledgments:** The authors express their gratitude to Professor P. S. Satsangi, for his kind inspiration and encouragement.

**Conflicts of Interest:** The authors declare no conflict of interest.

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
