# Peer review of "Optogenetic Generation of Neural Firing Patterns with Temporal Shaping of Light Pulses"

_photonics, doi:10.3390/photonics10050571_

Round 1

Reviewer 1 Report

In this study, the authors conducted a theoretical analysis of different waveforms of optogenetic light pulses (square, triangular, Gaussian, and Sinusoidal (12 subtypes)) in optogenetic excitation of hippocampal and cortical neurons. Three different opsins (Chronos, ChR2, and ChRmine) were examined and compared. The authors claimed that different light shapes produced different photocurrent amplitudes and kinetics, spike-timing, and spontaneous firing rate. Further, non-square waveforms generated more naturalistic spiking patterns compared to traditional square pulses.

The proposed strategy is interesting. However, the quality of the manuscript, the interpretation of the results, and the significance of the findings dampen the reviewer’s enthusiasm.  The reviewer was not convinced how these results provided significant information on developing different optogenetic strategies with different waveforms and opsins.

Main Comments

There were many confusions in this manuscript. Data was not well interpreted.

Just give some examples:

What were ISO, ISO-max, and ISO-Power? What was the amplitude for ISO-Max? What was pulse power? What amplitudes were used to match the pulse power?

There was no data showing the details of each waveform. What was the duration of ramping, what was the ramping angle, what were triangle waveform details, etc.?

It was hard to understand Figure 1, what is the x-axis? At which time point did the stimulation start? How long did photocurrent last?

Figure 1a, “At short light pulses, there is a slight difference in peak photocurrent and time to reach peak photocurrent on changing light pulse shape” What was the slight difference?

Figure 1b. very hard to understand what the author was trying to say. There was no stat comparison. “at long light pulse, the photocurrent kinetics significantly change from each other at different shapes” What was the difference? How did the authors conclude the significance?

The reviewer could not understand the conclusion “It is evident from Figure 1(a) that on illuminating with Gaussian light pulse, all three opsins do not show stable plateau current, which could be useful in smoothly turning-off of evoked firing in neurons”.

Figure 3a, there was no clear data interpretation. What is the x-axis of Figure 3c?

In Figures 4a and 5a, it was hard to follow what each color was for. The interpretation was confusing as well. 

Moderate editing 

Reviewer 2 Report

The manuscript “Optogenetic Generation of Neural Firing Patterns with Temporal Shaping of Light Pulses” by Dr. Himanshu Bansal et al presents an analysis of the temporal shaping of light pulses in optogenetic excitation of hippocampal neurons and neocortical fast-spiking interneurons expressed with ultrafast (Chronos), fast (ChR2), and slow (ChRmine) channelrhodopsins. Optogenetic excitation has been studied with light pulses of different temporal shapes that include square, forward / backward ramps, triangular, left / right triangular, Gaussian, left / right Gaussian, sinusoidal, and left / right sinusoidal. It was concluded that different light shapes result in significantly different photocurrent amplitudes and kinetics, spike-timing, and spontaneous firing rate. The optimal pulse width to achieve peak photocurrent for non-square pulses is 10 ms for Chronos, and 50 ms for ChR2 and ChRmine. Authors believe that obtained results show that non-square waveforms generate more naturalistic spiking patterns compared to traditional square pulses. Authors’ findings provide valuable insights for the development of new optogenetic strategies to better control neural activity in the brain.

 The authors have done a great job and I have no objections in fact, there are only some questions.

 Are the conclusions made by the author regarding the studied channelrhodopsin also valid for halorhodopsins? There is already at least one publication on halorhodopsins in the authors' literature review, I would suggest giving a few suggestions to optically sensitive chloride channels.

 Besides that the manuscript explores the relevant research literature and provides an   informative reference so I will be happy to recommend it for the publication after minor correction, suggested before.

Reviewer 3 Report

Thank you for inviting me to review this manuscript.

In this study the authors aim to present a detailed theoretical analysis of the temporal shaping of light pulses in optogenetic excitation of hippocampal neurons and neocortical fast-spiking interneurons expressed with ultrafast (Chronos), fast (ChR2), and slow (ChRmine) channelrhodopsins. 

This complex study is certainly interesting and worthy of being published. The figures and tables illustrate well the simulations carried out and the results.

I have just a few concerns:

  1. The study certainly has a fairly high degree of complexity, so it is not easy to describe it. However, in some points the discourse is difficult to follow and to understand. Could the authors try to make the text more fluid and understandable?

  2. Why all the simulations were implemented in MATLAB R2015a? We are currently on version 2023a. Could this be a problem to reproduce your work with newer matlab versions? Maybe it's time to include it in the limitations.

  3. The matlab script for the simulations has been stored in any repository?

  4. Does your work have limitations that could be included in the conclusions?

  5. emphasize better your strengths and future prospects.

Round 2

Reviewer 1 Report

None